

# Cyber-victimization and its association with depression among Vietnamese adolescents

Truc Thanh Thai[1], Mai Huynh Thi Duong[2], Duy Kim Vo[3], Ngan Thien Thi Dang[4], Quynh Ngoc Ho Huynh[1] and Huong Giang Nguyen Tran[5]

[1] Faculty of Public Health, University of Medicine and Pharmacy at Ho Chi Minh City, Ho Chi Minh City, Vietnam
[2] Pasteur Institute in Ho Chi Minh City, Ho Chi Minh City, Vietnam
[3] Long Dien District Medical Center, Ba Ria Vung Tau, Vietnam
[4] Pham Ngoc Thach hospital at Ho Chi Minh City, Ho Chi Minh City, Vietnam
[5] Training and Scientific Research Department, University Medical Center Ho Chi Minh City, Ho Chi Minh City, Vietnam

Corresponding authors
Truc Thanh Thai,
thaithanhtruc@ump.edu.vn
Huong Giang Nguyen Tran,
huong.tng@umc.edu.vn,
gianghuongtran07@gmail.com

## ABSTRACT

**Background.** Cyberbullying has become an alarming social issue, but little is known about its prevalence and consequences in many countries. This study investigated the prevalence of cyber-victimization and its association with depression among students in Ho Chi Minh City, Vietnam.

**Methods.** A cross-sectional study was conducted in 1,492 students from eight secondary schools and high schools in four urban and suburban areas. Multi-stage cluster sampling approach was used to recruit participants. Students participated in this study on a voluntary basis and completed a self-report questionnaire that included validated scales to measure their experience of cyber-bullying (Cyber Bullying Scale) and symptoms of depression (Center for Epidemiologic Studies–Depression Scale). Weighted logistic regression analysis was used to adjust for the cluster effect and sampling probability.

**Results.** Almost all (92.4%) students reported using the internet everyday and nearly 40% used internet for more than four hours per day. Cyber-victimization was identified in 36.5% of students and almost 25% experienced multiple types of cyber-victimization. Nearly half of students reported symptoms of depression. After adjusting for other covariates, students who experienced cyber-victimization were found to have 1.81 times (95% CI [1.42–2.30]) higher in odds of having symptoms of depression.

**Conclusions.** Cyber victimization and depression are both common in Vietnamese adolescents. Those who experienced cyber bullying have a higher likelihood of having symptoms of depression. These findings indicate an urgent need for interventions and policies targeting this emerging type of bullying in Vietnam and similar settings due to its potential harmful effects on adolescents' health.

## INTRODUCTION

Cyberbullying refers to any kinds of harassment through emails, chat rooms, websites or messages that inflict emotional distress on others (*Gladden et al., 2014*). In most countries, including Vietnam, there is an increasing trend in the number of internet users with the dominance of adolescents (*WeAreSocial, 2020*). In 2020, Vietnam is among the top 20 countries in the number of internet users and is estimated to reach more than 71 million users in 2021 (*Stats, 2021*; *Stats, 2021*). There is evidence that adolescents are at a high risk of becoming victim of cyberbullying (*Kowalski et al., 2014*; *Ruiz, 2019*). A meta-analysis of 131 studies evaluating the prevalence of cyberbullying from 2002 to 2013 revealed that 10%–40% of adolescents were cyber-victimized (*Kowalski et al., 2014*). A comparative analysis of existing reports in ASEAN countries found that the rate of cyber-victimization in adolescents has increased significantly in recent years (*Ruiz, 2019*). For example, in Philippines, this rate was especially high, up to 80% in 2015, although the government had enacted the Anti-Bullying Law against cyberbullying in 2013 (*Turner et al., 2013*). In Vietnam, a study conducted among secondary and high school students in northern provinces showed that approximately 81% of students had suffered from cyberbullying (*Le et al., 2017*).

Recently, cyberbullying has developed further into different types such as cancel culture, cyberstalking, flaming, trolling, masquerading and many others. Cyberbullying has become an alarming social issue not only because of its rising rate but also its serious physical and psychological consequences on adolescents (*Kowalski et al., 2014*; *Bottino, Bottino & Regina, 2015*). A systematic review revealed a relationship between cyber-victimization and moderate to severe symptoms of depression among 12–17-year-old students (*Bottino, Bottino & Regina, 2015*). For example, a study in Sweden among 1,214 students aged 13 to 16 found that students who experienced cyber victimization had up to 3.41 times higher in odds of having depression (*Landstedt & Persson, 2014*). Another study in the USA of 20,406 students in grade 6–12 reported that the rate of depressive symptoms was 2.61 times higher among cyber-bullying victims (*Schneider et al., 2012*). However, in Vietnam, research on this issue is still lacking and no study has been conducted in Ho Chi Minh City. Due to the lack of scientific evidence, interventions and policies against cyber bullying are also limited.

To date, there have been several studies conducted in Vietnam to evaluate depression and its potential risk factors. In 2011, a study of 1,161 students in Can Tho City showed that 41.1% students had depressive symptoms. Students who did not live with their parents, being physically or mentally abused, have poor academic performance and high study pressure were more likely to develop depressive symptoms (*Nguyen, Dedding & Pham, 2013*). Another study of 6,407 students found that 31.7% of students experienced depression. Noticeably, traditional bullying was found to be a significant risk factor of depression among students (*Tran, Le & Nguyen, 2020*). Ho Chi Minh City is one of the most important economic, political, cultural, and educational centers in Vietnam (*People's Committee of Ho Chi Minh City, 2021*). There is a syndesis in the prevalence of both internet use and depression in adolescents in this city with nearly 60% of students having symptoms

 

of depression (*Thai, Vu & Bui, 2020*). Apart from the known risk factors of depression, it is possible that there is an association between cyber-victimization and depression. If this hypothesis is true in Vietnam, as in other countries, more attention should be paid to internet-using habits and cyber-victimization because of their negative impacts on adolescents' health.

Therefore, the objective of this study was to investigate the prevalence of cyber-victimization and its association with depression among students in Ho Chi Minh City. The study serves as scientific background for further interventions for Vietnamese adolescents.

## MATERIALS & METHODS

### Settings and participants

A cross-sectional study was conducted in May 2020 in Ho Chi Minh City, Vietnam. At the time we conducted the study, there was no lockdown enforcement due to Covid-19 and all schools were open. The city had 272 public secondary schools and 110 public high schools across 24 districts (19 urban and 5 sub-urban). Multi-stage cluster sampling technique was used to recruit participants. We randomly selected four districts (three urban and one sub-urban), resulting in the selection of District 3, 5, Tan Phu and Hoc Mon. At each district, one secondary school and one high school were randomly selected from an exisiting list of schools provided by the Department of Education and Training. At each school, two classes in each grade were randomly selected from the list of classes. For secondary schools, only grade 8 and 9 were selected due to the anticipated potential information bias where young students might not fully understand the questions in the questionnaire. For high schools, all three grades (i.e., grade 10, 11, 12) were selected. A total of 40 classes from 4 secondary schools and 4 high schools were selected, including 8 classes from each grade. All students from the selected classes were invited to participate in the study.

The study protocol and all procedure conducted in this study were approved by the Ethical Committee for Biomedical Research at the University of Medicine and Pharmacy at Ho Chi Minh City, Vietnam (approval number: 97/ĐHYD-HĐĐĐ). Each selected class, including students and their parents, was informed about the study. Participation was voluntary and based on informed consent obtained from both the students and their parents or guardians. Among 1,648 students invited, 21 refused to participate and 129 students did not have consent from their parents or guardians, resulting in 1,498 students participating in this study. Among these, six students did not complete the questionnaire and were excluded from the analysis. The participation rate was 90.5% with 1,492 students in the analysis. All students completed a self-reported questionnaire which included information about their characteristics, internet use behaviors, living and studying environments, cyber-victimization and depression.

### Measurements

Students' characteristics measured in this study included sex, grade and grade point average in the last semester. Based on educational grading system in Vietnam, grade point average was categorized into low (<7/10), average/good ($\geq$7/10–8/10) and very

good/excellent (≥8/10). Questions about internet use behaviors included using internet every day (yes/no) and time spent on using internet (<2 hours/day, 2–4 hours/day and >4 hours/day). Internet addiction was measured using the Internet Addiction Test (IAT). The IAT has 20 items about the level of internet use using a Likert-type rating scale from 1 (never) to 5 (always) (*Chang & Law, 2008*; *Ni et al., 2009*). The total score ranges from 20 to 100 and was categorized into three levels, including no internet addiction (IAT score <50), mild/moderate internet addiction (IAT score of 50–79) and severe internet addiction (IAT score of 80 or more) (*Fatehi, Monajemi & Sadeghi, 2016*). The IAT demonstrates good psychometric properties in previous studies (*Milani, Osualdella & Blasio, 2009*; *Boysan, Kuss & Barut, 2017*) and has been used in Vietnamese adolescents (*Dang & Nguyen, 2013*; *Thai, Nguyen & Tran, 2018*).

Family environment was measured through Parental Bonding Instrument (PBI) (*Parker, Tupling & Brown, 1979*). The scale included 25 questions for both father and mother to measure their care and control. A higher total score indicates a higher level of care and control from parents. The PBI has been validated and used in a previous study in Vietnam (*Thai, Nguyen & Tran, 2018*). In this study, cutoffs from previous studies were used to dichotomize the scores including 24 for father control, 27 for mother control, 12.5 for father care and 13.5 for mother care. Information about the neighborhood environment included economic classification of living location (poor, average, rich), often experience fight, quarrel in the neighborhood (yes/no), often experience crime in the neighborhood (yes/no) and often witness violence among peers in the neighborhood (yes/no). School environment was evaluated using the School Connectedness Scale, which includes 5 Likert-type questions (*Furlong, O'brennan & You, 2011*). The total score ranges from 5 to 25, a higher score indicates a high level of school connectedness the students experienced. The scale has good psychometric properties in previous studies in Vietnamese adolescents (*Thai, 2010*; *Thai & Tran, 2016*).

The main variables were cyber victimization and symptoms of depression (SOD). Cyber victimization was measured through the 16-item Cyber Bullying Scale (CBS) (*Stewart, Drescher & Maack, 2014*). The CBS measures the levels and types of cyber victimization through a 5-point scale from 0 (never) to 4 (always). The experience of cyber victimization was identified when the students reported any level and type of cyber victimization in the past few months. The scale has a high level of internal consistency, construct and concurrent validity (*Stewart, Drescher & Maack, 2014*). SOD was measured using the Center for Epidemiologic Studies–Depression scale (CES-D) (*Radloff, 1991*). The CES-D includes 20 items asking about depressive symptoms in the past 7 days through a 4-point scale from 0 (never) to 3 (most of the times). The total score ranges from 0 to 60 and a higher score indicates a higher level of SOD. High levels of reliability and validity of the CES-D in comparison to various standards of depression diagnosis such as Mini-International Neuropsychiatric Interview (MINI), Composite International Diagnostic Interview (CIDI), and Structured Clinical Interview for DSM Disorder (SCID) were reported in previous studies (*Radloff, 1991*; *Vilagut et al., 2016*). Based on practice from previous studies among adolescents (*Thai et al., 2015*), a cutoff score of 16 was used to classify students with SOD.

### Data analysis

To take the advantage of sampling approach employed, the survey data analysis technique was used to account for the cluster effect and sampling weight. In this study, sampling weight was the inverse of the probability of students being recruited from the selected school. The sampling weight was used for further analyses including estimation through 95% confidence interval, statistical tests and logistic regression. Chi-squared tests were used to compare students' characteristics, living, studying environment and cyber victimization between students with SOD and students without SOD. To correct for the survey design, Chi-squared statistics were converted into F statistics. Variables with a *p* value of <0.1 were included in further analysis using univariate and multivariate logistic regression. Therefore, the association between cyber victimization and SOD was adjusted for other variables. The precision level was set at 5%. Statistical software Stata version 16 was used for all data analyses.

## RESULTS

Among 1,492 students in the survey-based analysis, most were females from high schools with an average/good grade point average in the last semester (Table 1). Almost all (92.4%, 95% CI [90.3%–94.1%]) students reported using the internet every day and nearly 40% used internet for more than four hours per day. Almost 60% of students had certain levels of internet addiction, including 57.9% mild/moderate (95% CI [54.2%–61.5%]) and 2.5% severe (95% CI [1.5%–3.9%]). Nearly half of students (47.8%, 95% CI [43.5%–52.1%]) reported SOD. Higher prevalence of SOD was found among female, high school students and those who spent more time on using internet and those who were classified as having internet addiction.

Table 2 presents students' living and studying environment, stratified by SOD. While about 40% of students reported caring from both father and mother, almost two-thirds disclosed control from them. Approximately 60% of students experienced fight, quarrel or crime in the neighborhood and one-third witnessed violence among peers in the neighborhood. Students had a moderate level of school connectedness. Cyber-victimization was identified in 36.5% of students (95% CI [32.9%–40.4%]) and almost 25% experienced multiple types of cyber-victimization. Students who received care from parents had a lower level of SOD. In contrast, those who reported control from parents experienced a high level of SOD. A higher level of SOD was also found among students experienced fight, quarrel, crime or witnessed violence among peers in the neighborhood. Students who had a lower level of school connectedness or experienced cyber-victimization were more likely to have SOD.

Subgroup analysis was performed for Tables 1 and 2 in males and females. Females who had a low grade point average, lived in a higher economical classification location and often experienced fight, quarrel or crime in the neighborhood were more likely to have SOD (Tables 1A and 2A). However, this pattern was not found in males. The associations between SOD and other factors were not different in males and females. The association between cyber-victimization and SOD remained statistically significant in both subgroups.

**Table 1 Characteristics and internet use behaviors among Vietnamese adolescents, stratified by symptoms of depression.**

| Characteristics | Total | Symptoms of depression | |
|---|---|---|---|
| | $n = 1,492$ n (Weighted %, 95% CI) | Yes $n = 673$, 47.8% (43.5–52.1) n (Weighted %, 95%CI) | No $n = 819$, 52.2% (47.9–56.5) n (Weighted %, 95%CI) |
| **Background information** | | | |
| **Sex** | | $\chi^2 (df = 1) = 13.7; F(1, 39) = 16.2; p < 0.001$ | |
| Female | 815 (55.3, 50.4–60.0) | 405 (60.2, 55.0–65.3) | 410 (50.7, 45.3–56.1) |
| Male | 677 (44.7, 40.0–49.6) | 268 (39.8, 34.7–45.0) | 409 (49.3, 43.9–54.7) |
| **Grade** | | $\chi^2 (df = 1) = 35.9; F(1, 39) = 37.2; p < 0.001$ | |
| High school | 892 (71.8, 56.2–83.5) | 460 (79.1, 65.1–88.5) | 432 (65.2, 48.4–78.8) |
| Secondary school | 600 (28.2, 16.5–43.8) | 213 (20.9, 11.5–34.9) | 387 (34.8, 21.2–51.6) |
| **Grade point average in the last semester** | | $\chi^2 (df = 2) = 2.9; F(2, 77.1) = 1.0; p = 0.363$ | |
| Very good/Excellent ($\geq 8/10$) | 517 (33.8, 26.9–41.5) | 218 (32.1, 24.9–40.3) | 299 (35.3, 27.7–43.7) |
| Average/Good ($\geq 7/10$-8/10) | 683 (48.1, 42.6–53.8) | 311 (48.3, 42.3–54.3) | 372 (48.0, 41.5–54.6) |
| Low (<7/10) | 292 (18.1, 13.8–23.4) | 144 (19.6, 14.8–25.5) | 148 (16.7, 11.9–22.8) |
| **Internet use behaviors** | | | |
| **Using internet every day** | | $\chi^2 (df = 1) = 3.2; F(1, 39) = 2.1; p = 0.151$ | |
| Yes | 1366 (92.4, 90.3–94.1) | 609 (91.1, 88.0–93.5) | 757 (93.6, 90.8–95.6) |
| No | 126 (7.6, 5.9–9.7) | 64 (8.9, 6.5–12.0) | 62 (6.4, 4.4–9.2) |
| **Time spent on using internet (hours/day)** | | $\chi^2 (df = 2) = 10.5; F(1.9, 74.7) = 5.3; p = 0.008$ | |
| <2 | 247 (15.5, 13.2–18.1) | 101 (14.2, 11.2–17.9) | 146 (16.6, 13.8–19.8) |
| 2–4 | 684 (46.6, 43.6–49.7) | 288 (43.6, 39.1–48.2) | 396 (49.4, 46.1–52.8) |
| >4 | 561 (37.9, 34.0–42.0) | 284 (42.2, 37.2–47.3) | 277 (34.0, 30.1–38.1) |
| **Internet addiction** | | $\chi^2 (df = 2) = 132.9; F(1.9, 74.2) = 56.8; p < 0.001$ | |
| No | 611 (39.6, 35.7–43.7) | 174 (25.2, 20.7–30.2) | 437 (52.9, 49.7–56.1) |
| Mild/moderate | 848 (57.9, 54.2–61.5) | 470 (70.3, 65.5–74.6) | 378 (46.6, 43.4–49.8) |
| Severe | 33 (2.5, 1.5–3.9) | 29 (4.6, 2.9–7.2) | 4 (0.5, 0.2–1.6) |

Significant variables in Tables 1 and 2 were analyzed in more details in Table 3. The association between cyber victimization and SOD was statistically significant in both univariate and multivariate analysis. The positive association between cyber victimization and SOD was confirmed after adjusting for other variables. Students who experienced cyber victimization had 1.81 times (95% CI [1.42–2.30]) higher in odds of having symptoms of depression. In addition, a higher likelihood of having SOD was also found among female students who had internet addiction, control from the mother. However, lower odds of having SOD was identified among students with caring from both mother and father and those with a high level of school connectedness.

## DISCUSSION

From a large pool of Vietnamese adolescents in this study, we found a very high prevalence of cyber victimization. Importantly, one-fourth of students experienced multiple types of cyber victimization. Despite using the same measuring scale (i.e., CBS), our figure is lower when compared with results obtained from the USA such as 58.7% in the study by *Stewart, Drescher & Maack (2014)* and 67.6% in the study by *Johnson (2016)*. Studies in other

Thai et al. (2022), *PeerJ*, DOI 10.7717/peerj.12907

**Table 2** Living and studying environment and cyber victimization among Vietnamese adolescents, stratified by symptoms of depression.

| Characteristics | Total | Symptoms of depression | |
|---|---|---|---|
| | $n = 1{,}492$ $n$ (Weighted %, 95% CI) | Yes $n = 673$, 47.8% (43.5–52.1) $n$ (Weighted %, 95% CI) | No $n = 819$, 52.2% (47.9–56.5) $n$ (Weighted %, 95% CI) |
| **Family environment** | | | |
| **Living with whom** | | $\chi^2 (df = 2) = 28.1; F(1.7, 66.1) = 12.8; p < 0.001$ | |
| With parents | 1275 (83.6, 79.8–86.9) | 543 (78.4, 73.0–83.0) | 732 (88.4, 84.9–91.1) |
| With either mother or father | 174 (13.2, 10.4–16.5) | 102 (16.9, 12.7–22.1) | 72 (9.7, 7.4–12.7) |
| With others | 43 (3.2, 2.4–4.2) | 28 (4.7, 3.5–6.2) | 15 (1.9, 1.2–3.0) |
| **Parental bonding** | | | |
| Caring from father | | $\chi^2 (df = 1) = 140.5; F(1, 39) = 128.9; p < 0.001$ | |
| | 658 (45.6, 41.1–50.2) | 179 (28.5, 24.8–32.5) | 479 (60.6, 55.1–65.7) |
| Control from father | | $\chi^2 (df = 1) = 28.7; F(1, 39) = 18.4; p < 0.001$ | |
| | 847 (62.7, 59.2–66.0) | 424 (70.2, 64.5–75.3) | 423 (56.1, 52.2–59.9) |
| Caring from mother | | $\chi^2 (df = 1) = 140.9; F(1, 39) = 88.0; p < 0.001$ | |
| | 574 (39.2, 35.2–43.3) | 137 (22.4, 17.9–27.6) | 437 (53.5, 49.1–57.9) |
| Control from mother | | $\chi^2 (df = 1) = 71.0; F(1, 39) = 64.1; p < 0.001$ | |
| | 863 (62.7, 59.3–66.0) | 448 (74.5, 69.7–78.8) | 415 (52.7, 49.1–56.2) |
| **Neighborhood environment** | | | |
| **Economical classification of living location** | | $\chi^2 (df = 2) = 4.5; F(2.0, 76.5) = 1.9; p = 0.152$ | |
| Poor | 46 (3.2, 2.4–4.4) | 26 (4.1, 2.5–6.5) | 20 (2.5, 1.7–3.7) |
| Average | 1362 (90.8, 88.7–92.6) | 602 (89.3, 86.0–91.8) | 760 (92.3, 89.8–94.2) |
| Rich | 84 (5.9, 4.7–7.5) | 45 (6.7, 5.0–8.9) | 39 (5.2, 3.7–7.4) |
| **Often experience fight, quarrel in the neighborhood** | | $\chi^2 (df = 1) = 8.2; F(1, 39) = 4.5; p = 0.041$ | |
| Yes | 948 (63.8, 60.7–66.7) | 455 (67.5, 62.7–71.9) | 493 (60.3, 55.9–64.6) |
| No | 544 (36.2, 33.3–39.3) | 218 (32.5, 28.1–37.3) | 326 (39.7, 35.4–44.1) |
| **Often experience crime in the neighborhood** | | $\chi^2 (df = 1) = 12.3; F(1, 39) = 5.6; p = 0.023$ | |
| Yes | 855 (58.8, 54.8–62.7) | 424 (63.5, 57.1–69.3) | 431 (54.5, 50.0–59.0) |
| No | 637 (41.2, 37.3–45.2) | 249 (36.5, 30.7–42.9) | 388 (45.5, 41.0–50.0) |

**Table 2** (*continued*)

| Characteristics | Total | Symptoms of depression | |
|---|---|---|---|
| | $n = 1{,}492$ $n$ (Weighted %, 95% CI) | Yes $n = 673$, 47.8% (43.5–52.1) $n$ (Weighted %, 95% CI) | No $n = 819$, 52.2% (47.9–56.5) $n$ (Weighted %, 95% CI) |
| **Often witness violence among peers in the neighborhood** | | $\chi^2 (df = 1) = 26.3; F(1, 39) = 23.6; p < 0.001$ | |
| Yes | 477 (33.8, 30.7–37.1) | 262 (40.4, 36.0–45.0) | 215 (27.8, 24.7–31.2) |
| No | 1015 (66.2, 62.9–69.3) | 411 (59.6, 55.0–64.0) | 604 (72.2, 68.8–75.3) |
| | **School environment** | | |
| **School connectedness score** (Mean & 95% CI) | | $F(1, 39) = 53.3; p < 0.001$ | |
| | 18.9 (18.5–19.3) | 17.9 (17.4–18.3) | 19.8 (19.4–20.2) |
| | **Cyber victimization** | | |
| **Cyber victimization** | | $\chi^2 (df = 1) = 84.3; F(1, 39) = 80.7; p < 0.001$ | |
| Yes | 528 (36.5, 32.9–40.4) | 321 (48.5, 43.3–53.7) | 207 (25.6, 22.2–29.3) |
| No | 964 (63.5, 59.6–67.1) | 352 (51.5, 46.3–56.7) | 612 (74.4, 70.7–77.8) |
| **Number of cyber-victimized forms experienced** | | $\chi^2 (df = 4) = 91.1; F(3.6, 141.4) = 21.7; p < 0.001$ | |
| 0 | 964 (63.5, 59.6–67.1) | 352 (51.5, 46.3–56.7) | 612 (74.4, 70.7–77.8) |
| 1 | 201 (12.8, 11.2–14.6) | 114 (15.3, 12.6–18.5) | 87 (10.5, 8.7–12.7) |
| 2 | 136 (9.5, 8.0–11.3) | 78 (12.4, 9.8–15.6) | 58 (6.9, 5.2–9.0) |
| 3 | 99 (7.0, 5.7–8.4) | 68 (10.3, 8.3–12.8) | 31 (3.9, 2.8–5.5) |
| 4+ | 92 (7.2, 5.4–9.7) | 61 (10.5, 7.9–13.8) | 31 (4.3, 2.7–6.7) |

**Table 3 Factors associated with symptoms of depression among Vietnamese adolescents (N = 1,492).**

| Characteristics | Symptoms of depression | | | |
| --- | --- | --- | --- | --- |
| | Crude | | Adjusted[a] | |
| | Weighted OR (95% CI) | *p* | Weighted OR (95% CI) | *p* |
| **Sex** *(female)* | 1.47 (1.21–1.79) | <0.001 | 1.80 (1.32–2.46) | <0.001 |
| **Grade** *(high school)* | 2.03 (1.60–2.57) | < 0.001 | 1.20 (0.94–1.54) | 0.143 |
| **Time spent on using internet (hours/day)** | | | | |
| <2 | 1 | | Ref | |
| 2–4 | 1.03 (0.73–1.44) | 0.875 | 0.72 (0.47–1.12) | 0.145 |
| >4 | 1.44 (1.05–1.98) | 0.024 | 1.13 (0.71–1.80) | 0.600 |
| **Internet addiction** | | | | |
| No | 1 | | Ref | |
| Mild/moderate | 3.17 (2.49–4.04) | <0.001 | 2.17 (1.67–2.82) | <0.001 |
| Severe | 18.04 (5.38–60.48) | <0.001 | 9.98 (3.71–26.81) | <0.001 |
| **Living with whom** | | | | |
| With parents | 1 | | Ref | |
| With either mother or father | 1.96 (1.32–2.92) | 0.001 | 1.52 (0.57–4.03) | 0.390 |
| With others | 2.81 (1.67–4.71) | <0.001 | 1.87 (0.48–7.39) | 0.358 |
| **Parental bonding** | | | | |
| Caring from father | 0.26 (0.20–0.33) | <0.001 | 0.51 (0.39–0.67) | <0.001 |
| Control from father | 1.84 (1.38–2.46) | <0.001 | 0.81 (0.57–1.16) | 0.243 |
| Caring from mother | 0.25 (0.18–0.34) | <0.001 | 0.48 (0.36–0.65) | <0.001 |
| Control from mother | 2.63 (2.05–3.37) | <0.001 | 2.04 (1.51–2.77) | <0.001 |
| **Often experience fight, quarrel in the neighborhood** | 1.36 (1.01–1.83) | 0.041 | 1.00 (0.67–1.51) | 0.988 |
| **Often experience crime in neighborhood** | 1.45 (1.06–1.99) | 0.023 | 1.04 (0.72–1.50) | 0.822 |
| **Experience violence among peers in the neighborhood** | 1.76 (1.39–2.23) | <0.001 | 1.01 (0.75–1.37) | 0.930 |
| **School connectedness score** | 0.86 (0.82–0.89) | <0.001 | 0.89 (0.86–0.93) | <0.001 |
| **Cyber victimization** | 2.74 (2.18–3.45) | <0.001 | 1.81 (1.42–2.30) | <0.001 |

Notes.
[a]Adjusted for other variables in the table.

countries also reported higher figures of cyber victimization prevalence such as Romania (37.3%) (*Athanasiou, Melegkovits & Andrie, 2018*) and China (44.5%) (*Rao, Wang & Pang, 2019*). Such differences can be attributed to the characteristic dissimilarity between regions regarding internet development and usage patterns (*Athanasiou, Melegkovits & Andrie, 2018*). Particularly, in our study, most cyberbullied students experienced rude comments (such as name-calling or being made fun of) through texting or online platforms. Meanwhile, in a study by Autry in 2016 on 543 US college students, the most common behavior was getting involved in online fights (*Autry, 2016*). Moreover, cyberbullying has still been a novel issue in developing countries like Vietnam for students, parents and even educators. However, the prevalence of cyber victimization in our study was relatively higher than previous reports in other regions of Vietnam, for example in Hue (9.0%) (*Nguyen, Nakamura & Seino, 2020*) and in northern provinces (24.0%) (*Le, 2020*).

Our study also unveiled an exceptionally high prevalence of depression among Vietnamese students, compared to other recent domestic studies. For example, depression

rates reported in two former studies in Vietnam by *Nguyen, Dedding & Pham (2013)* and *Tran, Le & Nguyen (2020)* were 41.1% and 37.1%, respectively. Regarding similar studies in the world on adolescents using the same measurement questionnaire, our study found an even higher prevalence. *Somrongthong, Wongchalee & Laosee (2013)* reported a depression rate in Thailand of 34.9% while *Li, Lau & Mo (2017)* found that about 23.5% of adolescents in China had symptoms of depression. All of these findings reveal that depression should be one of the priorities when planning public healthcare interventions for adolescents in all contexts across countries. To optimize the results of healthcare interventions, especially in under-resourced countries, one possible solution is to delineate the high-risk group of depression by identifying potential risk factors. Gender differences should also be considered in such interventions since females might be more vulnerable to SOD than males in some situations.

One of the unexplained issues in Vietnam is whether an association between cyberbullying and depression exists. After adjusting for other variables including important factors such as internet addiction and parental bonding, results from the final model in our study indicated that students who experienced cyber victimization had nearly two times higher the odds of having depression. This relationship has been confirmed by a large body of literature worldwide, irrespective of measuring scales of cyberbullying and depression; either evaluating by univariate or multivariate analysis; conducting in different countries with different cultures and timepoints. However, the level of association varies between regions, ranging from OR = 1.44 in the UK (*Fahy, Stansfeld & Smuk, 2016*) to OR = 2.70 in the US (*Alhajji, Bass & Dai, 2019*). In Asia, for example, a study in Taiwan found a 2.54 times higher in odds of having depression among those experienced cyber victimization (*Chang, Chiu & Miao, 2015*). In addition to the explanation through factors mentioned above, it might also be due to the dissimilar perception about cyber-victimization's severity or variance in the age distribution among different populations in previous studies (*Fahy, Stansfeld & Smuk, 2016*; *Nixon, 2014*). For example, *Agatston, Kowalski & Limber (2007)* found that teenage boys living in the USA were less likely to consider cyberbullying as a serious problem. As a result, the association between cyberbullying and depression might not be obvious. Moreover, it is encouraging to understand from our study that several factors such as school connectedness and parental bonding can protect the students against symptoms of depression although the mechanism and role of such factors warrant further investigation.

Findings from our study provide some major implications. Cyberbullying is an up-to-date issue overtime intrinsically and should be paid more attention, especially in developing countries where the internet era is thriving but the risks of interacting on cyber-space may not be fully perceived and understood. Cyberbullying has become more common, yet it received inadequate attention. What matters are that despite not being fully understood, cyberbullying already imposed harm on adolescent health and well-being in many ways, particularly mental health as found in this study and previous studies. Cyberbullying behaviors are being transferred into more sophisticated forms with relation to the development of technology such as the Internet, mobile phone and social networks and many others (*Khan et al., 2020*). Therefore, detailed description of cyberbullying forms may

provide a scientific reference for planning appropriate and effective intervention strategies. Additionally, although the effectiveness of legal regulations to control cyberbullying in other countries is unclear (*Foody et al., 2017*), many countries, including Vietnam, lack clear regulations on this issue. Overall, it is suggested that developing relevant policies and regulations are necessary to minimize possible risks in cyber-space. The present of such legislation and the use of reporting system can be beneficial so that cybervictims or their peers can report such cyberbullying cases. Importantly, the Covid-19 pandemic with unavoidable social distancing and lockdown in Vietnam and other countries provides hotbed for cyberbullying and depression among students due to distance learning (*Yang, 2021*). This warrants urgent actions to protect this vulnerable population.

Several limitations should be noted in this study. Due to the nature of the cross-sectional study the causal relationship cannot be confirmed. This is also a common limitation of the cross-sectional study design and thus additional follow-up studies should be conducted to determine the causal relationship between cyberbullying and depression. Moreover, this study was conducted in a large city in Vietnam, which is unrepresentative of the whole countries. It is likely that adolescents in other geographic areas, for example rural areas, may have different levels of using internet, different levels of depression, and thus the impact of cyberbullying on depression may be different. Therefore, more studies are needed in other provinces and regions in Vietnam. Additionally, more females participated in our study than males and thus might lead to potential biases. Although gender differences in particiation in mental health studies have been commonly reported in Vietnam (*Thai, Vu & Bui, 2020*; *Thai et al., 2020*; *Thai et al., 2021*), the reasons for this phenomenon and the magnitude of its effect on mental health and related factors warrant further investigations.

## CONCLUSIONS

Cyber victimization and depression are both common in Vietnamese adolescents. Those who experienced cyber bullying have a higher likelihood of having symptoms of depression. There is an urgent need for interventions and policies targeting this emerging type of bullying in Vietnam and similar settings due to its potential harmful effects on adolescents' health. More advance studies exploring the role of different types of cyberbullying and other mediators, moderators are also needed to better understand this association in different contexts.

## ACKNOWLEDGEMENTS

The authors would like to thank all students participating in this study. We also thank staff at participated schools for their support during data collection.

### Funding

This study received funding from the University of Medicine and Pharmacy at Ho Chi Minh City (177/2020/HD-DHYD). The funders had no role in study design, data collection and analysis, decision to publish, or preparation of the manuscript.

### Grant Disclosures

The following grant information was disclosed by the authors:
University of Medicine and Pharmacy at Ho Chi Minh City: 177/2020/HD-DHYD.

### Competing Interests

The authors declare there are no competing interests.

### Author Contributions

- Truc Thanh Thai conceived and designed the experiments, performed the experiments, analyzed the data, prepared figures and/or tables, authored or reviewed drafts of the paper, and approved the final draft.
- Mai Huynh Thi Duong and Duy Kim Vo conceived and designed the experiments, performed the experiments, authored or reviewed drafts of the paper, and approved the final draft.
- Ngan Thien Thi Dang analyzed the data, prepared figures and/or tables, and approved the final draft.
- Quynh Ngoc Ho Huynh analyzed the data, authored or reviewed drafts of the paper, and approved the final draft.
- Huong Giang Nguyen Tran conceived and designed the experiments, performed the experiments, prepared figures and/or tables, authored or reviewed drafts of the paper, and approved the final draft.

### Human Ethics

The following information was supplied relating to ethical approvals (i.e., approving body and any reference numbers):

Ethical Committee for Biomedical Research at the University of Medicine and Pharmacy at Ho Chi Minh City, Vietnam.

### Data Availability

The dataset and codebook are available in the Supplemental Files.

### Supplemental Information

Supplemental information for this article can be found online at http://dx.doi.org/10.7717/peerj.12907#supplemental-information.

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
