# Peer review of "Cyber-victimization and its association with depression among Vietnamese adolescents"

_PeerJ, doi:10.7717/peerj.12907_

## Round 0.1 · original submission · Major Revisions

Kindly add a specific objective of the research. If possible, add gender differences in the result section.

Reviewer 1 ·

Basic reporting

No comment

Experimental design

Not appropriate to stated objectives.

Validity of the findings

Valid.

Additional comments

1. Major concern I have with the paper is the language that the authors use which overstates the findings of their study. The (cross-sectional) design used by authors cannot support the objectives and conclusions drawn. The objectives are stated thus “to assess the effect of cybervictimization on depression” which implies a causal association. This is not supported by the design. The authors have to redo the language (say “association with” than “effect on”) everywhere in the paper, as appropriate.
2. Statistical analysis – better to say precision (in lieu of type I error) was set at 5%.
3. The study was carried out during the first wave of the COVID pandemic and students could have been spending more time on the internet for academic purpose, I would like to know how this was factored in when classifying IA as mild/moderate/severe based on parameters such as time spent using internet.

Reviewer 2 ·

Basic reporting

This manuscript is well-structured and in accordance with PeerJ standards. Clear and professional language is used; introduction and background show context with relevant literature being referenced; and figures are relevant, high quality, well-labelled and described.

The results section is very well-structured. First it demonstrates the relationship between symptoms of depression and internet use and internet addiction (Table 1). Second, it shows the association between symptoms of depression and the environment (family, neighborhood, school, and cyber victimization; Table 2). Finally in Table 3 the covariates are adjusted for, and the relationship between symptoms of depression and cyber victimization is reported.

In the supplementary materials Thanh Thai et al., have also included the raw dataset. As observed from the attached raw dataset, personally identifiable information is not present.

Minor comments on basic reporting
A sentence comparing the method used to measure symptoms of depression with DSM-5 needs to be made. This manuscript measured symptoms of depression using the Center for Epidemiologic Studies – Depression scale (lines 142–143). While the authors mention that this is a well-established assessment test, a sentence comparing/contrasting it with the standard DSM-5 is warranted.

This manuscript needs to discuss that there might be possible gender difference in self-reporting. That there were more female students volunteering for this study is also noteworthy (Table 1 and line 163). What is particularly interesting to note is that 1.5 times more female students self-reported symptoms of depression. Gender differences in self-reporting for epidemiological studies is well documented. This is particularly true for studies of mental health disorders, and this probable skew needs to be discussed.

The authors report chi-squared results (Tables 1, 2, and 3). The standard that the authors are following for reporting this statistic is not mentioned. It can be observed that this manuscript is conforming to the APA standard. However, reporting that fact would be ideal.

Experimental design

In this study Thanh Thai et al., aims to understand the relationship between symptoms of depression and cyber victimization of students in Vietnam. This original primary research article is within the scope of PeerJ. The authors showed rigor in defining their research question and designing experiments to address it.

Depression is a multifactorial psychiatric disorder. To control for this, the authors assessed for multiple categories at the level of individual student characteristics, environmental factors (family, neighborhood, school), and the main variables of cyber victimization and symptoms of depression. This shows necessary rigor. This is further highlighted by their sampling approach. Thanh Thai et al. utilized a multi-stage cluster sampling method. The authors selected secondary and higher secondary schools from 3 urban and 1 suburban area (grades 8–12) and invited all students from 40 classes of these schools to participate in this study. The measures used are well-established.

In addition, the authors have also shared the template for the participant information and consent form, questionnaire, and data codebook. This reviewer was able to easily load the provided raw data file and browse through it.

Validity of the findings

The data analysis methods — such as sampling weights — to control for the sampling approach is rigorous. Also, adjustment for confounding covariates was performed.

The authors also show balanced reporting of their conclusions. E.g., pertinent societal/household factors that can confer protection has been mentioned, and their findings are well contextualized with similar studies in other countries. In addition, the authors also dedicated an entire section in discussing the limitations of this study. It demonstrated that Thanh Thai et al., are mindful that this is not a causal study, and have been careful not to imply the same.

Additional comments

I recommend this study by Thanh Thai et al., for publication in PeerJ. Only minor comments on basic reporting.

---

## Round 0.2 · accepted · Accept

Thanks for your revised submission.

Reviewer 1 ·

Basic reporting

No concerns

Experimental design

No concerns

Validity of the findings

No concerns

Additional comments

Thank you for making the necessary changes.

Reviewer 2 ·

Basic reporting

The authors have addressed the comments.

Experimental design

NA

Validity of the findings

NA

Additional comments

NA